# Cytotoxic Cytochalasans from Sponge-Derived *Aspergillus* sp. SCSIO 41044

**DOI:** 10.3390/molecules30224483

**Published:** 2025-11-20

**Authors:** Xiaoyan Pang, Ini Wong, Qinlin Cao, Junfeng Wang, Xuefeng Zhou, Bin Yang, Junjian Wang, Hong Wang, Yonghong Liu

**Affiliations:** 1State Key Laboratory of Tropical Oceanography/Guangdong Key Laboratory of Marine Materia Medica, South China Sea Institute of Oceanology, Chinese Academy of Sciences, Guangzhou 510301, China; 2Sanya Institute of Marine Ecology and Engineering, Yazhou Scientific Bay, Sanya 572000, China; 3Guangdong Province Engineering Laboratory for Druggability and New Drug Evaluation, School of Pharmaceutical Sciences, Sun Yat-Sen University, Guangzhou 510006, China; 4University of Chinese Academy of Sciences, Beijing 100049, China

**Keywords:** sponge-derived fungus, *Aspergillus* sp. SCSIO 41044, cytochalasans, cytotoxicity

## Abstract

Two new cytochalasans (**1**–**2**), along with eight known ones (**3**–**10**) were isolated from sponge-derived *Aspergillus* sp. SCSIO 41044. The planar structures of **1**–**2** were elucidated through extensive spectroscopic analyses and their absolute configurations determined by modified Mosher’s methods. Biological evaluation revealed that **5**, **7**, and **8** showed potent cytotoxicity against small cell lung cancers H446 and H1048, with IC_50_ values ranging from 0.0441 to 1.61 μM.

## 1. Introduction

Cancer remains a major public health issue in China, with lung cancer ranking first in terms of both new cancer cases and cancer deaths in 2022 [1]. Also, according to global cancer survey data, lung cancer was the most commonly diagnosed cancer worldwide in 2022 [2]. Small cell lung cancer accounts for approximately 15% of all lung cancer cases. It is characterized by an extremely high rate of cell proliferation, a strong tendency to metastasize at an early stage, and an unfavorable prognosis. Despite extensive research on the mechanisms and drugs of small cell lung cancer, there is still a lack of small molecule drugs for treating small cell lung cancer in clinical practice [3,4]. Cytochalasans, the original meaning of which refers to the ability to disrupt actin filaments in cells, represent a category of fungal metabolites boasting diverse biological activities. These compounds always feature a unique tricyclic core structure fused to various aromatic or heteroaromatic rings, with the core consisting of an 11-, 13-, or 14-membered macrocyclic ring of polyketide origin [5,6]. Cytochalasans have demonstrated cytotoxic effects, and they are capable of inhibiting the growth and proliferation of cancer cells. Owing to this critical property, cytochalasans have attracted considerable attention in scientific research, particularly in the effort to develop them into potential anticancer agents [7,8,9]. Numerous structurally unique and biologically active secondary metabolites have been isolated from marine sources, with sponge-derived fungi being one of the most prolific [10,11]. In our ongoing investigations aimed at identifying chemically diverse secondary metabolites with anti-tumor activities, two new cytochalasans (**1**–**2**) and eight known ones (**3**–**10**) (Figure 1) were isolated from a liquid culture of sponge-derived *Aspergillus* sp. SCSIO 41044 under the guidance of Global Natural Products Social (GNPS) Molecular Networking analysis (Figure 2). Herein, we report the isolation, structural characterization, and anti-tumor activities of these cytochalasans.

## 2. Results and Discussion

Cytochalasin Z29 (**1**) was obtained as white powder, and was established as C_27_H_37_NO_6_ with ten degrees of unsaturation by a protonated molecule at *m*/*z* 472.2697 [M + H]^+^ in the HRESIMS spectrum and its ^13^C NMR data. Its ^1^H NMR spectrum (Table 1) showed five aromatic protons [δ_H_ 7.21–7.34, m, 5H], two sp^2^ methines [δ_H_ 5.65 (dd, J = 15.5, 9.5 Hz, CH-13), 5.31 (dd, J = 15.5, 7.0 Hz, CH-14)], one oxygenated sp^3^ methine [δ_H_ 3.62 (d, J = 7.0 Hz, CH-7)], one oxygenated sp^3^ methylene [δ_H_ 3.37–3.50 (m, CH_2_-20)], four methyls [δ_H_ 1.39 (s, CH_3_-11), 1.61 (s, CH_3_-12), 0.96 (d, J = 7.0 Hz, CH_3_-21), 1.18 (s, CH_3_-22)], and five active hydrogen protons [δ_H_ 7.80 (s, NH-2), 4.46 (t, J = 5.0 Hz, OH-3), 4.61 (d, J = 7.0 Hz, OH-7), 5.18 (s, OH-9), 5.08 (OH-17)]. Correspondingly, the ^13^C NMR and HSQC spectra (Table 1) displayed 27 carbons including two carbonyls (δ_C_ 218.6, 174.8), three sp^2^ nonprotonated carbons (δ_C_ 137.8, 131.3, 125.0), two oxygenated sp^3^ nonprotonated carbons (δ_C_ 77.9, 75.9), seven sp^2^ methines, five sp^3^ methines including one oxygenated methine (δ_C_ 70.7), four sp^3^ methylenes including one oxygenated methylene (δ_C_ 56.8), and four methyls (δ_C_ 25.8, 17.4, 16.8, 15.9).

Part of the NMR data of **1** showed similarity to those of cytochalasin Z13 (**9**) [12] which suggested **1** was a cytochalasin. The obvious differences were two more methylenes (CH_2_-19 and CH_2_-20) and one more oxygenated nonprotonated carbon (C-18) in **1** than in **9**, while oxygenated methine at C-18 in **9** was absent in the spectrum for **1**. The differences suggested **1** was a derivative of **9** with a 1-hydroxyethyl instead of a proton at C-18. The above suggestion was confirmed by correlations of H_2_-19/H_2_-20/OH-20 in the ^1^H-^1^H COSY spectrum and HMBC correlations from H_2_-19 to C-17, C-18, and C-22, from H_3_-22 to C-17, C-18, and C-19, from OH-18 to C-18, C-19, and C-22, and from OH-20 to C-19 (Figure 3). The planar structure of **1** was further established by its ^1^H-^1^H COSY spectrum and HMBC correlations (Figure 3).

The relative configuration of **1** was identified by analysis of the coupling constant data and NOESY spectra. The coupling constant of H-13 and H-14 is 15.5 Hz and the cross peak of H-8 and H-14 in NOESY spectrum indicated an *E* geometry of *Δ*^13^. The NOESY correlations of H_2_-10/H-4/OH-9/H-8/OH-7 indicated the same relative stereochemistry of the 3-benzyl-isoindol-1-one skeleton as the cytochalasans reported before [13,14,15]. The absolute configuration of C-7 was established as 7*S* by modified Mosher’s method [16,17] (Figure 4).

The molecular formula of cytochalasin Z30 (**2**) was assigned as C_28_H_37_NO_6_ with 11 degrees of unsaturation based on its HRESIMS spectrum and ^13^C NMR data. Its ^1^H NMR spectrum (Table 1) showed five aromatic protons [*δ*_H_ 7.20–7.34, m, 5H], four sp^2^ methines [*δ*_H_ 6.88 (dd, *J* = 15.5, 8.0 Hz, CH-19), 5.73 (d, *J* = 15.5 Hz, CH-20), 5.61 (dd, *J* = 15.5, 9.0 Hz, CH-13), 5.37 (dd, *J* = 15.5, 7.0 Hz, CH-14)], two oxygenated sp^3^ methines [*δ*_H_ 3.63 (brd, *J* = 6.5 Hz, CH-7), 3.13–3.18 (m, CH-17)], four methyls, and one active hydrogen proton [*δ*_H_ 7.77 (s, NH-2)]. Correspondingly, the ^13^C NMR and HSQC spectra (Table 1) displayed 28 carbons, including two carbonyls (*δ*_C_ 174.8, 167.8), three sp^2^ nonprotonated carbons (*δ*_C_ 137.8, 131.3, 125.0), one oxygenated sp^3^ nonprotonated carbon (*δ*_C_ 76.0), nine sp^2^ methines, seven sp^3^ methines including two oxygenated methines (*δ*_C_ 75.9, 70.9), two sp^3^ methylenes, and four methyls (*δ*_C_ 17.4, 17.0, 16.0, 13.7). Its NMR data closely resembled those of the first new cytochalasin reported by Miao et al. from *Aspergillus flavipes* RD-13 [15] except for the absence of methoxyl group at C-21 in compound **2**, which was further established by its ^1^H-^1^H COSY and HMBC correlations (Figure 3).

The coupling constants of H-13 and H-14, as well as of H-19 and H-20, are all 15.5 Hz, indicating the *E* geometry of *Δ*^13^ and *Δ*^19^, which was also supported by its NOESY correlations of H-8/H-14 and H-20/H-18. The NOESY cross peak of H-7/H-13 suggested that OH-7 and H-8 were in the same side. The signals of hydroxyl protons at C-7 and C-9 were not detected, and the signals of H-4 and H-8 overlapped in ^1^H NMR of **2**. The 1D NMR chemical shifts in C-1 to C-12 of **2** are nearly the same as those of **1,** with no more than 0.2 ppm offset (Table 1), which suggests that **2** also has same relative stereochemistry of the 3-benzyl-isoindol-1-one skeleton as **1**. The absolute configurations of C-7 and C-17 were established as 7*S* and 17*S* by Mosher’s modified method [16,17] (Figure 4).

The configuration of C-16 in the macrocyclic ring moieties was established as *S* in all cytochalasans isolated so far [7,18,19]. Open-chain cytochalasans are presumably biosynthesized from those cytochalasans with a 12-memberedmacrocyclic ring [13,20]. On this basis, we tentatively concluded that the two new cytochalasans (**1**–**2**) have an *S*-configuration at C-16. Therefore, the absolute configurations two new cytochalasans (**1**–**2**), except that of C-18, have been assigned as shown in Figure 1.

By comparing the ^1^H and ^13^C NMR and specific rotation data with the previous literature, seven known compounds were identified: 10-phenyl-[12]-cytochalasins Z8 (**4**) [18], 10-phenyl-[12]-cytochalasin Z7 (**5**) [18], cytochalasin Z17 (**6**) [20], 10-phenyl-[12]-cytochalasin Z16 (**7**) [14], *Δ*^6,12^-isomer of 5,6-dehydro-7-hydroxy-cytochalasin E (**8**) [21], cytochalasin Z13 (**9**) [13], and cytochalasin Z11 (**10**) [13]. It is worth mentioning the configuration of *Δ*^18^ for compound **3**; its 1D NMR data were similar to those of cytochalasin Z17 (**6**) reported by Lin et al. [20], except for C-17(*δ*_C_ 211.1) and C-23 (*δ*_C_ 19.0) in **3** with obvious shifts downfield, as well as C-19 (*δ*_C_ 124.3) and C-20 (*δ*_C_ 32.1) in **3** with obvious shifts upfield in comparison with those in **6** (Appendix A), while the COSY and HMBC correlations of **3** (Appendix A) suggested the same planar structure with **6**. The differences in chemical shifts may cause by the different configuration of *Δ*^18^ [22], which was also established as *Z* based on the cross peak of H-19 and H_3_-23 in NOESY spectrum. However, cytochalasin Z17 bearing a *Z* configuration at C-18 has been reported by Zhang et al. [14], and upon careful reading of this article, it was discovered that the name and structure of cytochalasin Z17, which in the article is described as the *Z* configuration at C-18, are inconsistent with the *E* configuration shown by its X-ray crystallographic drawing. Thus, compound **3** was identified as 18*Z*-cytochalasin Z17.

All isolated compounds (**1**–**10**) were assessed for their cytotoxicity against the small cell lung cancer cell lines H446 and H1048. As a result, compounds **3**–**8** exhibited varying degrees of inhibitory effect, among which compound **5**, **7**, and **8** showed significant cytotoxicity with IC_50_ values ranging from 0.044 to 1.61 μM (Figure 5). Structurally, compounds **3**–**8** all possess a 12-/13-membered macrocyclic fragment, while compounds **5**, **7**, and **8** all have a double bond between C-6 and C-12. Thus, the 12-/13-membered macrocyclic fragment and the double bond between C-6 and C-12 may be essential for the cytotoxic activity of cytochalasans. Compounds **1**–**10** were also tested for their antimicrobial activities against three plant-pathogenic fungi (*Alternaria alternate*, *Curvularia australiensis*, and *Rhizoctonia solani*), as well as three human pathogenic bacteria (*Staphylococcus aureus*, methicillin-resistant *Staphylococcus aureus*, and *Exiguobacterium profundum*). But none of those compounds showed observable activity with 50 μg per 6 mm disk.

## 3. Materials and Methods

### 3.1. General Experimental Procedures

HRESIMS spectra were recorded on a Bruker maXis Q-TOF mass spectrometer in positive ion mode. One-dimensional and two-dimensional NMR spectra were measured on a Bruker AV 500 or 700 MHz NMR spectrometer with TMS as an internal standard. Optical rotations were measured using an MCP-500 Polarimeter (Anton, Graz, Austria). ECD and UV spectra were measured with a circular dichroism spectrometer (Applied Photophysics). MPLC was carried on SepaBean machine with YMC ODS-A (12 nm, S-50 μm YMC). HPLC was carried on Hitachi primaide with YMC ODS SERIES (YMC-Pack ODS-A, YMC Co., Ltd., Kyoto, Japan, 250 × 10 mm I.D., S-5 μm, 12 nm). Chromatographic pure organic solvents were purchased from DWSCI HiPurSolv Co., Ltd. Wuhan, China. (*S*)-α-methoxy-α-phenylacetic acid (MPA) and (*R*)-MPA were purchased from J&K Scientific, Beijing, China. The analytical reagents (EtOAc and MeOH) were purchased from Jindongtianzheng Fine Chemical Reagent Factory, Tianjin, China. The deuterated reagents were purchased from Qingdao Tenglong Microwave Technology Co., Ltd., Qingdao, China.

### 3.2. Fungal Material

The fungal strain SCSIO 41044 was obtained from the sponge collected from Yitong Shoal, Hainan Province, China. The producing strain was stored on MB agar (malt extract 15 g, sea salt 10 g, agar 16 g, H_2_O 1 L, and pH 7.4–7.8) slants at 4 °C and deposited at the Guangdong Key Laboratory of Marine Materia Medica, South China Sea Institute of Oceanology. The ITS1-5.8S-ITS2 sequence region (541 base pairs, accession number PX239440) of strain SCSIO 41044 was amplified by PCR. The Web BLAST (https://blast.ncbi.nlm.nih.gov/Blast.cgi, accessed on 9 October 2025) analysis against the GenBank database revealed that its DNA sequence shared 99% similarity with *Aspergillus urmiensis*. Thus, the strain was identified as *Aspergillus* sp.

### 3.3. Fermentation, Extraction, and Isolation

The mass fermentation of this fungus was carried out in 19.8 L liquid medium (10 g soluble starch, 1 g tryptone, 20 g sea salt, and 1 L tap water), which was divided into 66 flasks, with each containing 300 mL, at 25 °C under static conditions. After 30 days, the cultures were soaked in EtOAc (500 mL/flask), and the mycelia were cut into small pieces and sonicated for 20 min. The EtOAc solution was concentrated under reduced pressure to gain a crude extract (9.3 g).

The crude extract was subjected to reversed-phase C-18 MPLC eluted with MeOH/H_2_O (10:90–100:0, *v*/*v*) and separated into seven fractions (Fr-1–Fr-7). Fr-6 (2.8 g) was applied to reversed-phase C-18 MPLC again eluted with MeOH/H_2_O (30:70–100:0, *v*/*v*) to obtain five sub-fractions (Fr-6-1–Fr-6-6). Fr-6-1 was subjected to semipreparative HPLC (32% CH_3_CN/H_2_O, 2 mL/min) to gain **10** (9.5 mg, *t*_R_ = 42.0 min) and **1** (5.7 mg, *t*_R_ = 45.0 min). Fr-6-2 was subjected to semipreparative HPLC (72% CH_3_OH/H_2_O, 2 mL/min) to obtain **2** (6.4 mg, *t*_R_ = 16.5 min). Fr-6-3 was subjected to semipreparative HPLC (60% CH_3_OH/H_2_O, 2 mL/min) to afford **8** (12.6 mg, *t*_R_ = 23.5 min) and **9** (13.7 mg, *t*_R_ = 19.5 min). Fr-6-5 was subjected to semipreparative HPLC (72% CH_3_OH/H_2_O, 2 mL/min) to gain **3** (14.1 mg, *t*_R_ = 17.0 min) and **6** (38.2 mg, *t*_R_ = 18.0 min). Fr-7 (1.1 g) was subjected to semipreparative HPLC (2 mL/min) to obtain **7** (12.8 mg, 58% CH_3_CN/H_2_O, *t*_R_ = 15.2 min), **5** (8.0 mg, 75% CH_3_OH/H_2_O, *t*_R_ = 13.8 min) and **4** (22.1 mg, 75% CH_3_OH/H_2_O, *t*_R_ = 14.5 min).

### 3.4. Structural Characterizations of ***1***, ***2*** and ***3***

*Cytochalasin Z29* (**1**): white powder; [*α*]^20^_D_ + 54.8 (*c* 0.10, MeOH); UV (MeOH) *λ*max (log*ε*) 200 (3.62) nm; ECD (0.35 mM, MeOH) *λ*max (*Δε*) 202 (34.75), 219 (0.11), 222 (2.44) and 236 (−0.61) nm; ^1^H and ^13^C NMR data, Table 1; HRESIMS *m*/*z* 472.2697 [M + H]^+^ (calcd for C_27_H_38_NO_6_, 472.2694).

*Cytochalasin Z30* (**2**): white powder; [*α*]^20^_D_ + 47.4 (*c* 0.10, MeOH); UV (MeOH) *λ*max (log*ε*) 200 (3.56) nm; ECD (0.68 mM, MeOH) *λ*max (*Δε*) 202 (20.09), 218 (−0.87), 222 (1.12) and 236 (−0.36) nm; ^1^H and ^13^C NMR data, Table 1; HRESIMS *m*/*z* 484.2692 [M + H]^+^ (calcd for C_28_H_38_NO_6_, 484.2694).

18*Z-cytochalasin Z17* (**3**): white powder; [*α*]^20^_D_ + 81.4 (*c* 0.10, MeOH); UV (MeOH) *λ*max (log*ε*) 200 (3.39), and 239 (2.44) nm; ECD (0.71 mM, MeOH) *λ*max (*Δε*) 200 (25.50), and 233 (−4.47) nm; ^1^H and ^13^C NMR data, Appendix A; HRESIMS *m*/*z* 464.2442 [M + H]^+^ (calcd for C_28_H_34_NO_5_, 464.2431).

### 3.5. Preparation of the (S)- and (R)-MPA Esters of Cytochalasins Z29 and Z30 Using the Modified Mosher’s Method

Cytochalasins Z29 (**1**) (1.0 mg) and a small mixing rotor were added into a round-bottom flask (5 mL) and dried in a vacuum desiccator for 2 h. Then, dimethylaminopyridine (0.1 mg), dicyclohexylcarbodiimide (0.1 mg), (*R*)-MPA (0.69 mg), and 500 μL of CD_2_Cl_2_ were added in sequence and the reaction mixture was stirred for 16 h at room temperature. Then the crude product was purified by semipreparative HPLC (80% CH_3_OH/H_2_O, 2.2 mL/min, 10.5 min) to yield the (*R*)-MPA ester **1a** (0.7 mg): white powder; ^1^H NMR (DMSO-*d*_6_, 700 MHz) 5.19 (brd, *J* = 7.0 Hz, H-7), 2.31 (t, *J* = 9.1 Hz, H-8), 1.32 (brs, H_3_-11), 1.24 (s, H_3_-12), 5.75 (dd, *J* = 15.4, 9.1 Hz, H-13), 4.94 (ddd, *J* = 19.6, 9.1, 4.9 Hz, H-14), 1.87 (m, H-15a), 1.56 (dt, *J* = 13.3, 9.1 Hz, H-15b), 3.12 (brdd, *J* = 14.0, 7.0 Hz, H-16), 0.88 (s, H_3_-21); ESIMS: *m*/*z* 790.3 [M + Na]^+^. By the same procedure, the (*S*)-MPA ester **1b** (0.8 mg): white powder; ^1^H NMR (DMSO-*d*_6_, 700 MHz) 5.13 (brd, *J* = 8.4 Hz, H-7), 2.40 (t, *J* = 8.4 Hz, H-8), 1.18 (brs, H_3_-11), 1.12 (s, H_3_-12), 5.89 (dd, *J* = 15.4, 9.8 Hz, H-13), 5.30 (ddd, *J* = 15.4, 7.7, 6.3 Hz, H-14), 2.18 (m, H-15a), 1.84 (dt, *J* = 14.0, 8.4 Hz, H-15b), 3.24 (brdd, *J* = 14.0, 7.0 Hz, H-16), 0.98 (s, H_3_-21); ESIMS: *m*/*z* 790.3 [M + Na]^+^ was obtained from the reaction of **1** (1.0 mg) with (*S*)-MPA (0.69 mg).

Cytochalasins Z30 (**2**) (1.0 mg) and a small mixing rotor were added into a round-bottom flask (5 mL) and dried in a vacuum desiccator for 2 h. Then, dimethylaminopyridine (0.15 mg), dicyclohexylcarbodiimide (0.26 mg), (*R*)-MPA (2.75 mg), and 500 μL of CD_2_Cl_2_ were added in sequence and the reaction mixture was stirred for 16 h at room temperature. Then, the crude product was purified by semipreparative HPLC (80% CH_3_OH/H_2_O with 0.04% formic acid, 2.2 mL/min, 23.5 min) to yield the (*R*)-MPA ester **2a** (0.9 mg): white powder; ^1^H NMR (DMSO-*d*_6_, 500 MHz) 5.17 (brd, *J* = 8.0 Hz, H-7), 2.35 (m, H-8), 1.33 (s, H_3_-11), 1.26 (s, H_3_-12), 5.74 (dd, *J* = 15.5, 9.5 Hz, H-13), 4.98 (dt, *J* = 15.5, 6.5 Hz, H-14), 1.67 (m, H-15a), 1.31 (m, H-15b), 1.55 (m, H-16), 4.66 (t, *J* = 6.0 Hz, H-17), 2.54 (m, H-18), 6.46 (dd, *J* = 15.5, 8.5 Hz, H-19), 5.56 (d, *J* = 15.0 Hz, H_2_-20), 0.67 (d, *J* = 6.5 Hz, H_3_-22), 0.67 (d, *J* = 6.5 Hz, H_3_-23); ESIMS: *m*/*z* 778.3 [M − H]^−^. By the same procedure, the (*S*)-MPA ester **2b** (0.9 mg): white powder; ^1^H NMR (DMSO-*d*_6_, 500 MHz) 5.07 (brd, *J* = 8.0 Hz, H-7), 2.38 (m, H-8), 0.89 (s, H_3_-11), 1.22 (s, H_3_-12), 5.59 (dd, *J* = 15.5, 10.0 Hz, H-13), 5.19 (dt, *J* = 15.0, 6.5 Hz, H-14), 1.64 (m, H-15a), 1.26 (m, H-15b), 1.67 (m, H-16), 4.71 (dd, *J* = 7.0, 4.5 Hz, H-17), 2.62 (m, H-18), 6.49 (m, H-19), 5.77 (d, *J* = 15.5 Hz, H_2_-20), 0.54 (d, *J* = 6.5 Hz, H_3_-22), 0.94 (d, *J* = 6.5 Hz, H_3_-23); ESIMS: *m*/*z* 778.2 [M − H]^−^ was obtained from the reaction of **2** (1.0 mg) with (*S*)-MPA (2.75 mg).

### 3.6. Cytotoxicity Assay

H446 and H1048 cells were purchased from the American Type Culture Collection (ATCC). Cells were seeded in 96-well plates at 5000–8000 cells per well, with each well containing a total volume of 100 μL of media. After 24 h, compounds were serially diluted and added 50 μL to the cells per well. Following four days of incubation. CCK-8 reagents (Selleck chemicals, America) were added, and the 96-well plates were incubated at 37 °C in 5% CO_2_ incubators for 1–2 h. The absorbance at 450 nm was measured using a microplate reader (Promega) according to the manufacturer’s instructions. The final results were expressed as percentages relative to the vehicle-treated cell group, which was defined as 100% [23,24].

## 4. Conclusions

In summary, under the guidance of GNPS molecular network analysis two new cytochalasans (**1**–**2**), along with eight known ones (**3**–**10**) were isolated from sponge-derived *Aspergillus* sp. SCSIO 41044. The modified Mosher’s method was employed to determine the absolute configurations of compounds **1** and **2**. Compounds **5**, **7**, and **8** showed potent cytotoxicity against small cell lung cancers H446 and H1048, with IC_50_ values ranging from 0.044 to 1.61 μM. Analysis of the structure–activity relationship reveals that the 12-/13-membered macrocyclic fragment and the double bond between C-6 and C-12 may be essential for the cytotoxic activity of cytochalasans. The present study enriches the structural diversity of cytochalasans, and may lay a foundation for the discovery of anti-tumor marine drugs.

## Figures and Tables

**Figure 1 molecules-30-04483-f001:**
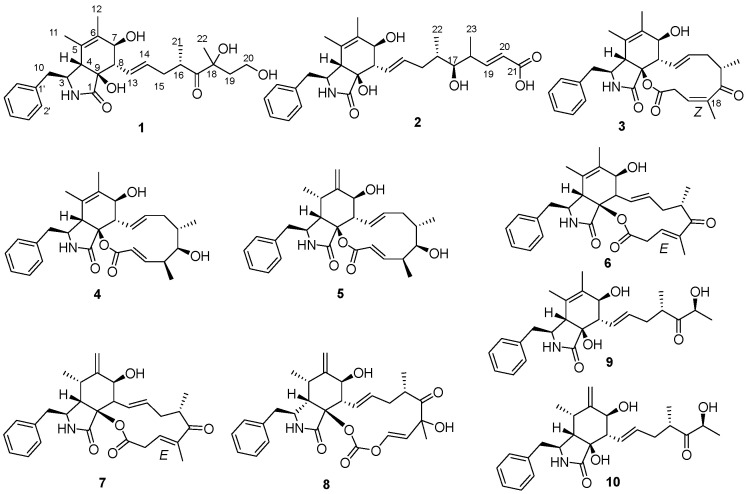
Cytochalasans (**1**–**10**) from sponge-derived *Aspergillus* sp. SCSIO 41044.

**Figure 2 molecules-30-04483-f002:**
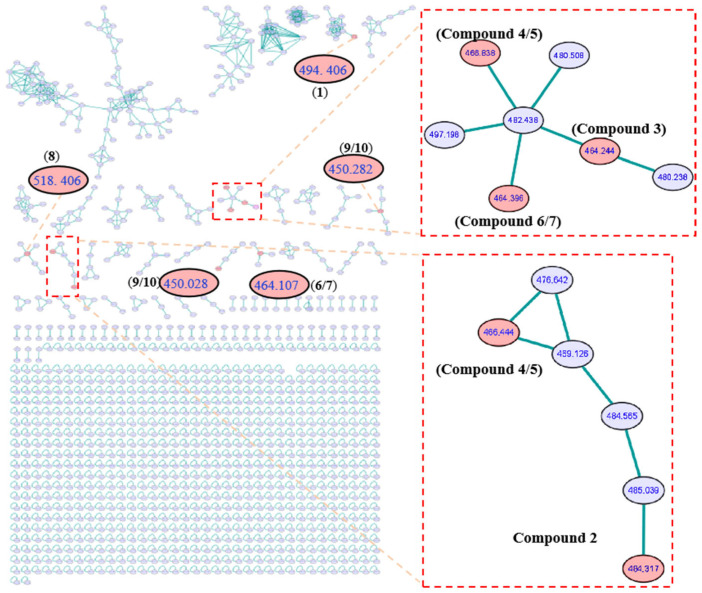
GNPS molecular network obtained through HR-ESI-MS/MS analysis of extracts from *Aspergillus* sp. SCSIO 41044 and clusters corresponding to cytochalasans observed in this molecular network.

**Figure 3 molecules-30-04483-f003:**
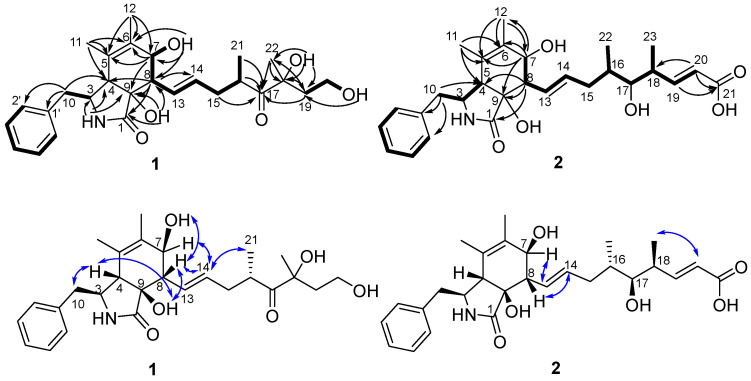
Key ^1^H-^1^H COSY (**－**), HMBC (→), and NOESY (↔) correlations of compounds **1**–**2**.

**Figure 4 molecules-30-04483-f004:**
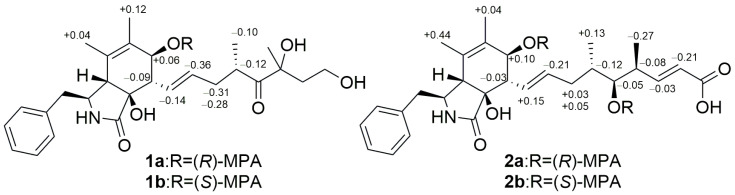
*Δδ* value [*Δδ* (in ppm) = *δ*_R_–*δ*_S_] obtained for (*R*)-MPA and (*S*)-MPA esters of compounds **1** and **2**.

**Figure 5 molecules-30-04483-f005:**
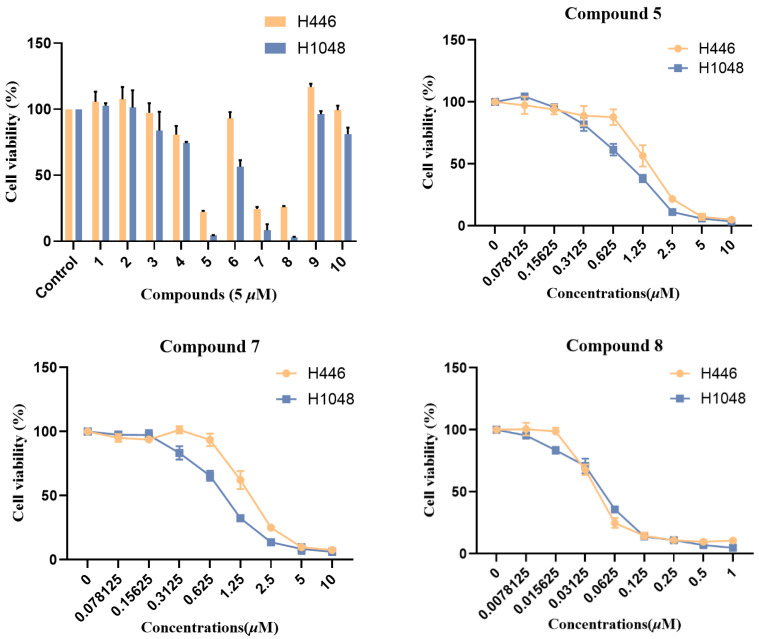
Cytotoxic activities of isolated compounds (**1**–**10**).

**Table 1 molecules-30-04483-t001:** NMR data for compounds **1**–**2** (500/125 MHz, TMS, *δ* ppm) in DMSO-*d*_6_.

Position	1		2	
	*δ*_C_, type	*δ*_H_, mult, *J*	*δ*_C_, type	*δ*_H_, mult, *J*
1	174.8 C		174.8 C	
2 (NH)		7.80, s		7.77, s
3	56.8 CH	3.29–3.34, m	56.7 CH	3.30–3.34, m
4	51.9 CH	2.33, m	51.9 CH	2.30–2.35, m
5	125.0 C		125.0 C	
6	131.3 C		131.3 C	
7	70.7 CH	3.62, t, 7.0	70.9 CH	3.63, brd, 6.5
8	52.4 CH	2.29, dd, 9.0, 7.0	52.3 CH	2.30–2.35, m
9	75.9 C		76.0 C	
10	42.2 CH_2_	3.01, dd, 13.0, 7.0 2.88, dd, 13.5, 5.5	42.2 CH_2_	3.02, dd, 13.0, 7.0 2.88, dd, 13.0, 5.0
11	17.4 CH_3_	1.39, s	17.4 CH_3_	1.41, s
12	15.9 CH_3_	1.61, s	16.0 CH_3_	1.62, s
13	129.0 CH	5.65, dd, 15.5, 9.5	128.2 CH	5.61, dd, 15.5, 9.0
14	131.4 CH	5.31, dt, 15.5, 7.0	132.4 CH	5.37, dt, 15.5, 7.0
15	36.0 CH_2_	2.22, brdt, 13.5, 5.5 1.88, dt, 13.5, 8.0	36.8 CH_2_	2.04, dt, 13.5, 5.5 1.83, dt, 14.0, 8.0
16	38.8 CH	3.25, dq, 13.5, 6.5	35.9 CH	1.47–1.55, m
17	218.6 C		75.9 CH	3.13–3.18, m
18	77.9 C		39.5 CH	2.42, dt, 14.0, 7.0
19	41.8 CH_2_	1.83, ddd, 14.0,8.0, 6.5 1.65, ddd, 8.0, 6.0	151.8 CH	6.88, dd, 15.5, 8.0
20	56.8 CH_2_	3.37–3.50, m	121.8 CH	5.73, d, 15.5
21	16.8 CH_3_	0.96, d, 7.0	167.8 C	
22	25.8 CH_3_	1.18, s	13.7 CH_3_	0.80, d, 7.0
23			17.0 CH_3_	0.95, d, 6.5
1′	137.8 C		137.7 C	
2′,6′	129.8 CH	7.21–7.27, m	129.8 CH	7.20–7.27, m
3′,5′	128.3 CH	7.29–7.34, m	128.3 CH	7.29–7.34, m
4’	126.4 CH	7.21–7.27, m	126.4 CH	7.20–7.27, m
OH-7		4.61, d, 7.0		
OH-9		5.08, s		
OH-18		5.18, s		
OH-20		4.46, t, 5.0		

## Data Availability

The authors declare that the data supporting the findings of this study are available within the article and Appendix A.

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
