# Peer review of "Cytotoxic Cytochalasans from Sponge-Derived Aspergillus sp. SCSIO 41044"

_molecules, 2025, doi:10.3390/molecules30224483_

Round 1

Reviewer 1 Report

Comments and Suggestions for Authors

Please see attached

Comments on the Quality of English Language

This manuscript would be greatly improved by being reviewed and edited by an English language speaker

Author Response

Response to the comments of Reviewer #1:

This manuscript describes the isolation and cytotoxicity of a suite of structurally related cytochalasan natural products isolated from a sponge derived Aspergillus fungal species. Three of the compounds are novel, and are elucidated primarily through comparison with previously reported published data and NMR analysis. The absolute configuration of two of the new compounds was determined by chiral shift NMR analysis using a modified Mosher’s method. Overall, the structural determinations are sound, although there are a few important areas that will need to be clarified, as well as some mistakes in the chemical shift data. The grammar and text mistakes throughout should be reviewed by a native English speaker– just a few representative examples will be provided below. For the cytotoxicity determinations, the method used should be stated in the text (in this case, a CCK-8 assay) and the EC50s should be determined from dose response curves with a fitted line from linear regression analysis.

Page 2,line 44-45 “chemical diversity” should be “chemically diverse”

Answer:Thanks very much for your suggestion. The phrase of “chemical diversity” has been revised to “chemically diverse” and  change has been marked in the revised manuscript with blue highlight.

Page 4, line 75 “9 was disappeared “ should be “was absent in the spectrum for 1

Answer:Thanks very much for your suggestion.The sentence has been revised as you suggested and marked with blue highlight in the revised manuscript.

page 4, line 76 “compound of 9 that a 1-hydroxyethyl replaced the proton at C-18” should be “derivative of 9 with a 1-hydroxyethyl group instead of a proton at C-18”

Answer:Thanks very much for your suggestion. The sentence has been revised as you suggested and marked with blue highlight in the revised manuscript.

page 4, line 84 “by the coupling constant and NOESY spectrum” should be “by analysis of the coupling constant data and NOESY spectra”.

Answer:Thanks very much for your suggestion. The sentence has been revised as you suggested and marked with blue highlight in the revised manuscript.

page 5, line 96 “5.73 (d, J = 15.5” should have Hz after 15.5

Answer: Thanks very much for your careful review. “Hz” has been added after 15.5 and marked with blue highlight in the revised manuscript.

page 5, line 107-114 –this whole section needs to be edited for grammar and tense. (many more instances throughout…..)

Answer: Thanks very much for your suggestion. Grammar and tense in this whole section have been checked and revised. Changes have been marked with blue highlight in the revised manuscript.

page 5, line 104–this should say Miao et al. instead of S. Miao

Answer: Thanks very much for your suggestion. The description of S. Miao has been revised to Miao et al and marked with blue highlight in the revised manuscript.

page 5, lines 112-113. The same chemical shifts for the two compounds is not convincing evidence for assuming that they share the same relative configurations.

Answer: Thanks very much for your careful review. Because signals of hydroxyl protons at C-7 and C-9 were not detected, and the signals of H-4 and H-8 overlapped in 1H NMR of 2, only cross-peak of H-7/H-13 was obtained in its NOESY spectrum. The relative configuration of the 3-benzyl-isoindol-1-one skeleton in 2 was established by comparing chemical shifts. This method is based on the description of two new seco-cytochalasins in the previous reference by Miao et al (The Journal of Antibiotics (2022) 75:410-414).

For the Mosher’s analysis of compounds 1 and 2, there should be some discussion about the multiple hydroxyl groups in the molecules and how only the MTPA monoesters were isolated.

Answer: Thanks very much for your careful review. I am sorry for my careless. The R at C-20 of compound 1 in Figure 4 was missing before, and which has been added. In the esterification between new compounds (1 and 2 ) and MPA, an excess of MPA was added, with two equivalents added to each reaction site. The hydroxyl groups on the quaternary carbon usually do not participate in the reaction, and the added MPA is excessive, so primary hydroxyl groups and secondary hydroxyl groups have both reacted completely.

Page 5, line 119. All carbon shifts should only have a single digit after the decimal. (ie. 19.01 should be 19.0)

Answer: Thanks very much for your careful review. 19.01 has been revised to 19.0 and all carbon shifts has been checked carefully. Changes have been marked in the revised manuscript with blue highlight.

Page 5, line 119. This section is confusing and some of the designations seem wrong. For example, the C17 and C-23 signals for 3 are shifted downfield compared to 6 (it says upfield in the text). Line 120: the chemical shift for C-19 is incorrect here in the text compared to the table.

Answer: Thanks very much for your careful review. As you said, the description of this sentence was wrong, and words “downfield” and “upfield” were used upside down. The sentence and the chemical shift for C-19 have been revised and marked in the revised manuscript with blue highlight.

Page 5, line 124. The NOESY correlations between H3 and H10 would not indicate the relative stereochemistry for this center.

Answer: Thanks very much for your question. I am very sorry for my negligence of missing other key NOESY correlations. Compound 3 has been described as a known compound, and its description has been simplified.

Page 6. For the cytotoxicity data, it might be worthwhile to include even a brief discussion about any structure activity relationships since a suite of related compounds were tested.

Answer: Thanks very much for your suggestion. A brief discussion about structure activity relationships has been added at the ending of Part 2 and marked in the revised manuscript with blue highlight.

Page 6, methods

There is no section describing the source and brands of any chemical compounds or solvents.

Answer: Thanks very much for your careful review. Information of analytical reagents, chromatographic pure organic solvents, deuterated reagents and chemical reagents used for esterification of Mosher’s modified method were added in the part of 3.1.

page 6, line 158. After the accession number, this sentence needs to be fixed

Answer: Thanks very much for your careful review. This sentence has been rewritten and marked with blue highlight in the revised manuscript.

page 7, line 166. Was the extract filtered or decanted before concentration?

Answer: Thanks very much for your question. After sonicated and settling the ethyl acetate phase and aqueous phase separated into layers, so the concentration of the ethyl acetate phase was not filtered.

page 7, line 190–this section does not match the data presented in the text (the mass and formula are completely different)

Answer: Thanks very much for your careful review. The mass and formula of 3 in text was wrong before. Now compound 3 has been described as a known compound, and its description has been simplified.

Page 7, line 197. The phrase “drained the water with vacuum dryer” is unclear to me-I don’t know what this means.

Answer: Thanks very much for your question. In the modified Mosher's method, an anhydrous environment must be maintained during the esterification reaction. Thus, compound 1 must be dried in a vacuum desiccator for 2h prior to the reaction. The expression of this sentence is confusing and has been revised and marked with blue highlight in the revised manuscript.

Page 7, line 197. Reagents with abbreviations should be written out at least the first time. I don’t know what an R-MPA is–should this say R-MTPA? A citation should be added here for the method used.

Answer: Thanks very much for your question. R-MPA is the abbreviation for (R)-α-methoxy-α-phenylacetic acid, which has been added in the part of 3.1. Two articles related to the modified Mosher’s method (Chem. Rev. 2004, 104, 17−117. & J. Nat. Prod. 2021, 84, 1993−2003) have been cited in the text. Changes have been marked with blue highlight in the revised manuscript.

Page 8, line 232. The brand for the CCK-8 kit should be added.

Answer: Thanks very much for your suggestion. The CCK-8 kit was purchased from Selleck chemicals of America. This information has been added and marked with blue highlight in the revised manuscript.

page 8, line 238. Instead of “to sum up”, this could say “In summary”.

Answer: Thanks very much for your suggestion. The phrase of “to sum up” has been revised to “In summary” and change has been marked in the revised manuscript with blue highlight.

Reviewer 2 Report

Comments and Suggestions for Authors

The authors isolated analogs of cytochalasans from Aspergillus and elucidated their structures. Sturcutural elucidation of analogs is important.

This result is worthy of publication in Molecules.

The determination of the planar structure is clear by the NMR analyses and in comparison with previous analogs and the determination of the stereo configuration is based mainly on biosynthesis.

1. MPA should describe the compound name.

2. The literature on the modified Mosher method is missing.

3. There is a question about the Δδ distribution of compound 2 by the modified Mosher method.

Both C7 and C17 hydroxy groups are the beta direction.

However, the Δδ distribution of around 7-OH and 17-OH appears to be reversed.

The Δδ for H11 and H12 on the right side of 7-OH is positive and that is negative for H8. On the other hand, the Δδ  of H18 and H23 on the right side of the 17-OH is negative and that is positive for H16 and H22.

Please check the NMR data.

4. No mention is made of the lack of cytotoxicity of compounds 1-3.

Author Response

Response to the comments of Reviewer #2:

The authors isolated analogs of cytochalasans from Aspergillus and elucidated their structures. Sturcutural elucidation of analogs is important.

This result is worthy of publication in Molecules.

The determination of the planar structure is clear by the NMR analyses and in comparison with previous analogs and the determination of the stereo configuration is based mainly on biosynthesis.

  1. MPA should describe the compound name.

Answer: Thanks very much for your suggestion. MPA is the abbreviation for α-methoxy-α-phenylacetic acid, which has been added in the part of 3.1. Change has been marked in the revised manuscript with blue highlight.

  1. The literature on the modified Mosher method is missing.

Answer: Thanks very much for your careful review. Two articles related to the modified Mosher’s method (Chem. Rev. 2004, 104, 17−117. & J. Nat. Prod. 2021, 84, 1993−2003) have been cited in the text. Changes have been marked with blue highlight in the revised manuscript.

  1. There is a question about the Δδdistribution of compound 2by the modified Mosher method.

Both C7 and C17 hydroxy groups are the beta direction.

However, the Δδ distribution of around 7-OH and 17-OH appears to be reversed.

The Δδ for H11 and H12 on the right side of 7-OH is positive and that is negative for H8. On the other hand, the Δδ of H18 and H23 on the right side of the 17-OH is negative and that is positive for H16 and H22.

Please check the NMR data.

Answer: Thanks very much for your question. NMR data of Mosher esters for 2 have been checked carefully, and no errors was found. The absolute configurations of OH-7 and OH-17 were all established as S referring to the model as shown below reported in previous literature (Chem. Rev. 2004, 104, 17−117). This literature has been cited in the text and changes have have been marked with blue highlight in the revised manuscript.

Reviewer 3 Report

Comments and Suggestions for Authors

The manuscript describes the isolation, structural elucidation and a preliminary biological evaluation of 10 cytochalasans isolated from the sponge-derived Aspergillus sp. SCSIO 41044, of which three compounds (1-3) were identified as new cytochalasan derivatives. The structural elucidation of these three new metabolites was performed by NMR techniques, including 1D and 2D experiments, and by HRMS analyses. In addition, the Mosher's method was employed for the determination of the absolute configurations of the compounds 1 and 2. These structural elucidations were correctly achieved and provided consistent structures, featuring a brominated pyrrole framework in their structures. The biological evaluation of all these 10 compounds was achieved by analysis of their cytotoxic activities against two cancer lines, revealing that only 3 compounds displayed significant cytotoxicities. According to these results, the manuscript can be considered for publication after the corresponding revision according to the following comments: 

1. According to the authors, compound 3 is resported as a new cytochalasan. However, it is possible to find in the literature that this compound has already been isolated and described. In particular, this compound has been described in Planta Med. 2010, 76, 1616 and Nat. Prod. Res. 2019, 33, 2939. However, comparing the NMR spectroscopic data of the isolated compound 3 with those reported in the aforementioned articles, it is possible to conclude that there is no match between them, so they seem to correspond to different products. Therefore, the structural elucidation carried out in these articles, including this one, could be incorrect. The authors must mention these articles and justify whether this compound 3 is indeed new.

2. The 3 new compounds are described as white powders. The authors should include their corresponding melting points. 

3. The biological evaluation of the isolated compounds is considered very modest. Since the cytochalasans exhibit a broad range of biological activities, including antitumor, antimicrobial, anti-inflammatory, and immunosuppressive activities, I suggest that the authors include a more detailed and comprehensive biological evaluation of the new compounds, not limited to their cytotoxic properties but extended to other activities, such as antibiotic or antifungi activities.

Author Response

Response to the comments of Reviewer #3:

The manuscript describes the isolation, structural elucidation and a preliminary biological evaluation of 10 cytochalasans isolated from the sponge-derived Aspergillus sp. SCSIO 41044, of which three compounds (1-3) were identified as new cytochalasan derivatives. The structural elucidation of these three new metabolites was performed by NMR techniques, including 1D and 2D experiments, and by HRMS analyses. In addition, the Mosher's method was employed for the determination of the absolute configurations of the compounds 1 and 2. These structural elucidations were correctly achieved and provided consistent structures, featuring a brominated pyrrole framework in their structures. The biological evaluation of all these 10 compounds was achieved by analysis of their cytotoxic activities against two cancer lines, revealing that only 3 compounds displayed significant cytotoxicities. According to these results, the manuscript can be considered for publication after the corresponding revision according to the following comments: 

  1. According to the authors, compound 3is resported as a new cytochalasan. However, it is possible to find in the literature that this compound has already been isolated and described. In particular, this compound has been described in Planta Med. 2010, 76, 1616 and Nat. Prod. Res. 2019, 33, 2939. However, comparing the NMR spectroscopic data of the isolated compound 3 with those reported in the aforementioned articles, it is possible to conclude that there is no match between them, so they seem to correspond to different products. Therefore, the structural elucidation carried out in these articles, including this one, could be incorrect. The authors must mention these articles and justify whether this compound 3 is indeed new.

Answer: Thanks very much for your careful review. I am very sorry for my negligence of missing this two literature you mentioned. Compound 3 in this article cannot be reported as a new compound. Compound 3 has been described as a known compound in the revised manuscript. NMR and X-ray data of cytochalasin Z17 reported by Zhang et al (Planta Med. 2010, 76, 1616) indicated that the configuration of Δ18 was E but was named as Z. The X-ray Cytochalasin Z17 was reported by Akhter et al (Nat. Prod. Res. 2019, 33, 2939) as a known compound and no NMR data was found. Thus, compound 3 in our manuscript was identified as 18Z-cytochalasin Z17.

  1. The 3 new compounds are described as white powders. The authors should include their corresponding melting points. 

Answer: Thanks very much for your suggestion. The two new compounds (1-2) underwent esterification of modified Mosher’s method, antitumor and antibacterial experiments, after that only a few amount remained adhering to the bottom of the bottle as shown below, and it was no longer in the form of a distinct solid powder, making it impossible to determine their melting points.

  1. The biological evaluation of the isolated compounds is considered very modest. Since the cytochalasans exhibit a broad range of biological activities, including antitumor, antimicrobial, anti-inflammatory, and immunosuppressive activities, I suggest that the authors include a more detailed and comprehensive biological evaluation of the new compounds, not limited to their cytotoxic properties but extended to other activities, such as antibiotic or antifungi activities.

Answer: Thanks very much for your suggestion. Acturally, antimicrobial bioassays of all isolated compounds (1-10) in this manuscript against three human pathogenic bacteria (Staphylococcus aureus, methicillin-resistant Staphylococcus aureus, Exiguobacterium profundum) and three plant-pathogenic fungi (Alternaria alternate, Curvularia australiensis, and Rhizoctonia solani) have been tested before. But these compounds didn’t shown observable activity with 50 μg per 6mm disc. The original antibacterial activity photos and the corresponding relationship of compound numbers between those in the manuscript and in the photos are shown below. As your opinion, the results of the antibacterial experiment has been added and described briefly in the revised manuscript.

Compound 1 corresponds to H16; Compound 2 corresponds to H18;

Compound 3 corresponds to H14; Compound 4 corresponds to H5;

Compound 5 corresponds to H6; Compound 6 corresponds to H15;

Compound 7 corresponds to H8; Compound 8 corresponds to H9;

Compound 9 corresponds to H10; Compound 10 corresponds to H17

Round 2

Reviewer 3 Report

Comments and Suggestions for Authors

The authors must revise the paragraph in page 5, lines 128-136. There are some unfinished sentences that must be corrected. In addition, they should explain better whether compound 3 finally matches with the reported in the literature. The authors mention in their response that X-ray of Cytochalasin Z17 was reported by Akhter et al (Nat. Prod. Res. 2019, 33, 2939). However, in this article X-ray of Z17 is not reported. Therefore, the authors should explain this issue more clearly and in a more detailed manner before final acceptance.

Author Response

The authors must revise the paragraph in page 5, lines 128-136. There are some unfinished sentences that must be corrected. In addition, they should explain better whether compound 3 finally matches with the reported in the literature. The authors mention in their response that X-ray of Cytochalasin Z17 was reported by Akhter et al (Nat. Prod. Res. 2019, 33, 2939). However, in this article X-ray of Z17 is not reported. Therefore, the authors should explain this issue more clearly and in a more detailed manner before final acceptance.

Answer: Thanks very much for your suggestion. As your opinion, sentences you mentioned at lines 128-136 have been revised and described clearly in the revised manuscript and marked with blue highlight.

By the way, sentence of “The X-ray Cytochalasin Z17 was reported by Akhter et al (Nat. Prod. Res. 2019, 33, 2939) as a known compound and no NMR data was found.” in the previous response was written incorrectly. I forgot to erase this two words “The X-ray”. I just wanted to say “ Cytochalasin Z17 was reported by Akhter et al (Nat. Prod. Res. 2019, 33, 2939) as a known compound and no NMR data was found.”. I sincerely apologize for the confusion caused by my carelessness.
